# A system for inducible mitochondria-specific protein degradation in vivo

Swastika Sanyal [1] ✉, Anna Kouznetsova [2], Lena Ström[2] & Camilla Björkegren [2] ✉

Targeted protein degradation systems developed for eukaryotes employ cytoplasmic machineries to perform proteolysis. This has prevented mitochondria-specific analysis of proteins that localize to multiple locations, for example, the mitochondria and the nucleus. Here, we present an inducible mitochondria-specific protein degradation system in *Saccharomyces cerevisiae* based on the *Mesoplasma florum* Lon (mf-Lon) protease and its corresponding ssrA tag (called PDT). We show that mitochondrially targeted mf-Lon protease efficiently and selectively degrades a PDT-tagged reporter protein localized to the mitochondrial matrix. The degradation can be induced by depleting adenine from the medium, and tuned by altering the promoter strength of the *MF-LON* gene. We furthermore demonstrate that mf-Lon specifically degrades endogenous, PDT-tagged mitochondrial proteins. Finally, we show that mf-Lon-dependent PDT degradation can also be achieved in human mitochondria. In summary, this system provides an efficient tool to selectively analyze the mitochondrial function of dually localized proteins.

Originating from alpha-proteobacteria, mitochondria have their own genome[1,2]. Approximately 1% of the mitochondrial proteome is expressed by this genome, while the rest is nuclear-encoded, translated by cytoplasmic ribosomes, and guided by mitochondrial targeting signals (MTS) to their final location in mitochondria[3,4]. However, at least a third of the mitoproteome is localized to additional organelles[5]. These dually localized proteins, for example those functioning in both the nucleus and mitochondria[6–9], generally score poorly for the features that define conventional MTSs[10], which prevents analysis of their mitochondria-specific functions by MTS disruption.

So far, a conditional mitochondria-specific protein degradation system has been lacking, as most eukaryotic degron systems employ host cell's degradation machinery, resulting in a complete depletion of the protein of interest by the cytosolic ubiquitin-proteasome system [e.g., AID (Auxin induced degron)][11]. Most investigations employ complete or partial depletion of the dually localized proteins which does not allow analysis of their mitochondrial functions exclusively[6–8,12]. Other methods for depleting mitochondrial

proteins, such as in vitro silencing of genes present in mitochondrial chromosomes, have been recently reported[13]. While an excellent method for inhibition of mitochondrial genome expression, it is limited to the 1% of the mitoproteome that is encoded by the mitochondrial genome. Furthermore, being an in vitro system, it lacks applicability when it comes to the analysis of mitochondrial dynamics, which is intricately linked with other organelles[14,15]. Therefore, the development of a method for mitochondria-specific protein degradation is crucial.

To achieve this, we sought to engineer a system to degrade proteins specifically inside the mitochondria. Given the prokaryotic origins of mitochondria, we borrowed from the so-called ribosome rescue mechanism, ubiquitously present in bacteria but absent from mitochondria[16]. In this process, a degradation-inducing signal (called ssrA tag) is added at the end of a translating mRNA lacking a STOP codon. This allows the release of the aberrant mRNA and subsequent proteolysis of the partially translated protein by endogenous proteases[17]. We used the ssrA-degradation system from a simple mollicute, *Mesoplasma florum* (mf), which employs its Lon protease

[1]Karolinska Institutet, Department of Biosciences and Nutrition, Neo, Hälsövägen 7c, 141 83 Huddinge, Sweden. [2]Karolinska Institutet, Department of Cell and Molecular Biology, Biomedicum, Tomtebodavägen 16, 171 77 Stockholm, Sweden. ✉e-mail: swastika.sanyal@ki.se; camilla.bjorkegren@ki.se

as the sole ssrA-degrading protease[18]. We combined the mf-Lon protease with a variant of the mf-ssrA epitope called protein degradation tag [PDT, developed by Cameron and Collins[19], see Supplementary Notes], which allows a more faithful recognition by the mf-Lon protease. Using mf-Lon and the PDT tag, we have developed a mitochondria-specific protein degradation system that functions both in yeast and human cells.

## Results

### Mesoplasma florum Lon protease degrades PDT-tagged GFP in yeast mitochondria

To test if mf-Lon can perform PDT-dependent degradation inside yeast mitochondria we created yeast strains co-expressing mitochondrially targeted mf-Lon (mito-mf-Lon) and green fluorescent protein (GFP), either unmodified (mito-GFP), or tagged with PDT (mito-GFP-PDT)

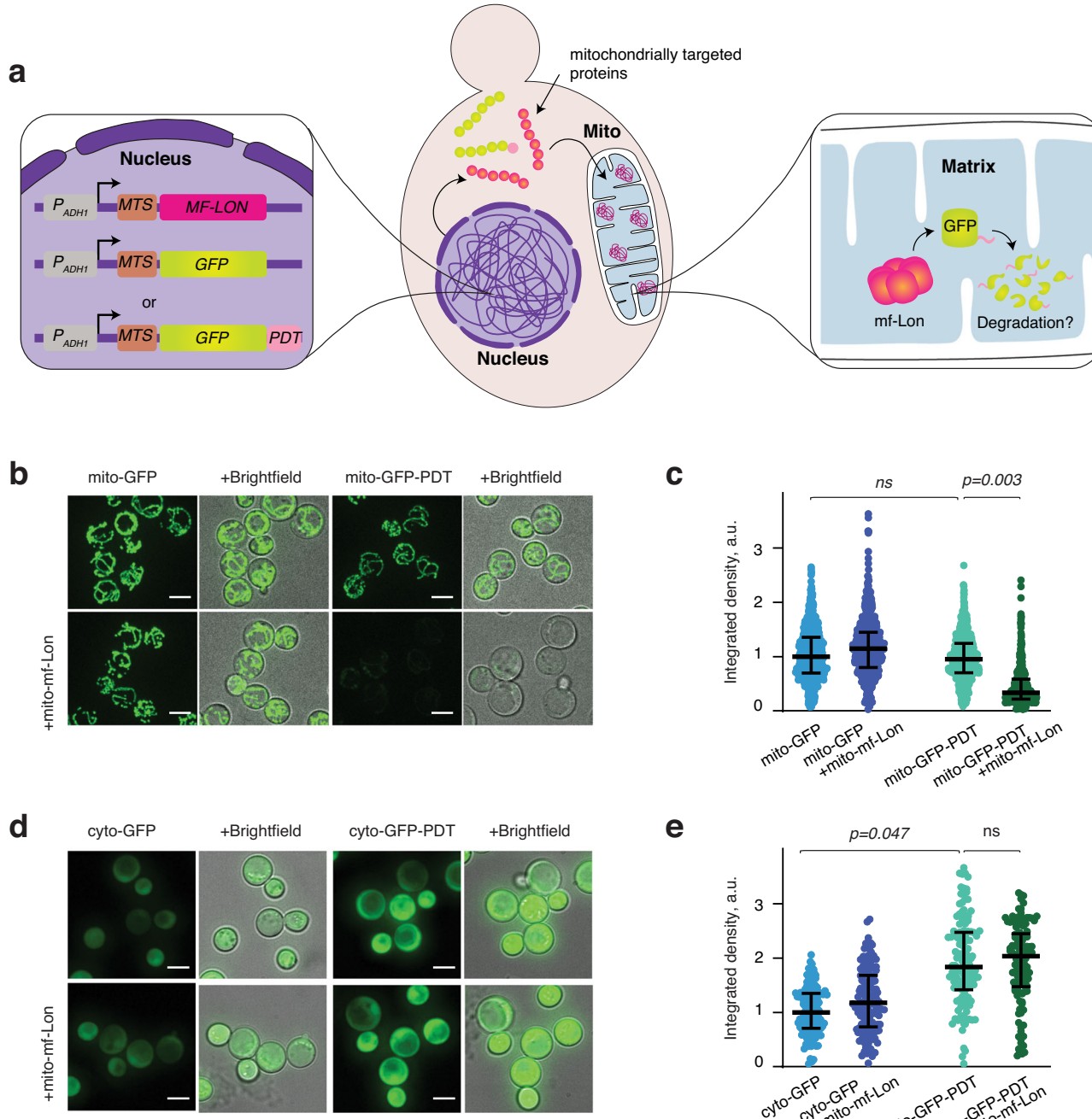

**Fig. 1 | Mitochondria-specific degradation of GFP-PDT by *M. florum* Lon protease. a** Schematics showing experimental set up. **b** Representative live cell images of yeast cells expressing mitochondrially-targeted GFP or GFP-PDT, either independently (upper panels) or together with mitochondrially-targeted *M. florum* Lon (lower panels). Cells were grown in the standard minimal medium for -16–18 hours before sample preparation. Scale bar = 5 μm. **c** Dot plots showing quantification of GFP intensities as arbitrary units (a.u.) of the indicated yeast strains, with each dot representing an individual cell. Data depict median value with interquartile range and normalized to the mito-GFP strain. Pairwise comparisons were made by hierarchical resampling[53], and a *P*-value (*p*) <0.05 was considered significant (ns = not significant). At least 600 cells from each sample were collected over five independent experiments. **d** Similar to **b**, but strains with GFP lacking the MTS are shown. **e** Similar to **c**, except that dot plots show GFP intensities of cells in **d**, and data are normalized to the cyto-GFP strain. At least 100 cells from each sample were collected over two independent experiments.

(Fig. 1a). This revealed a significant loss of the GFP signal in the mito-chondria of cells which expressed both mitochondrially-targeted GFP-PDT and mf-Lon, but not in the absence of mf-Lon, or when GFP lacked the PDT tag or the MTS (Fig. 1b–e, Supplementary Fig 1a–c). Cyto-plasmically localized GFP-PDT signal was instead significantly stronger than its untagged counterpart (Fig. 1d, e). The loss of mito-GFP-PDT signal in mito-mf-Lon expressing cells was robust, and more than 80% of the cells displayed an acute depletion of the signal (Supplementary Fig. 1a, b). Together, this shows that mf-Lon can detect and specifically degrade PDT-tagged GFP inside mitochondria.

Since respiration is dispensable in yeast, and the mitochondrial volume and metabolism changes during different growth phases and carbon sources[20,21], we investigated if PDT degradation required specific growth conditions. We found that mf-Lon-induced mito-GFP-PDT degradation did not occur in undefined rich medium (Fig. 2a). However, when a culture initiated in rich medium was transferred to synthetically defined minimal medium, the GFP-PDT signal was strongly reduced in mf-Lon expressing cells after over-night growth (16 h–20 h) (Fig. 2b). To temporally determine when GFP-PDT was degraded during the overnight incubation period, we followed the GFP signal in mf-Lon cells co-expressing mito-GFP, or mito-GFP-PDT (Fig. 2c). GFP-PDT degradation was initiated when cells entered deceleration growth phase, ~5.5 hours after inocula-tion at an optical density ($OD_{600}$) of 0.2 and continued to decrease during prolonged cell growth (Fig. 2d, e). Contrarily, the signal of mitochondrial GFP lacking PDT increased during and after cellular growth slowed down (Fig. 2e). These data demonstrate that the mitochondrial GFP-PDT degradation by mf-Lon occurs after cells have entered deceleration growth phase in synthetic minimal medium.

### Adenine depletion induces mf-Lon-dependent GFP-PDT degradation

Yeast cell duplication decelerates when glucose, supporting rapid fermentative growth, becomes limiting. Under such conditions, cells undergo the metabolic reprogramming needed for respiration of ethanol, which is the fermentation product of glucose[20,22]. This tran-sition from fermentation to respiration is known as the diauxic shift and involves catabolite de-repression and is accompanied by an increase in mitochondrial DNA (mtDNA)[23,24]. Moreover, in yeast strains that are defective in adenine production, depletion of the base pre-cedes glucose exhaustion, the diauxic shift is thereby reached earlier, and growth slows down even in the presence of sufficient extracellular glucose[25].

Taking advantage of the fact that the strain used for this inves-tigation is an adenine auxotroph, we tested whether adenine can be used to control GFP-PDT degradation. Supplementing standard minimal medium with 50 mg/l adenine prevented GFP-PDT degra-dation (Supplementary Fig. 2a), establishing that adenine depletion is central to the degradation in post-diauxic cells. To test this directly, we resuspended cells growing in an excess of adenine in medium containing 0, 2, or 20 mg/l adenine, and at a glucose concentration of 1%, which is similar to the extracellular glucose concentration when cells reach diauxie[20,25] (Fig. 3a). The mitochondrial GFP-PDT signal in mf-Lon expressing cells was reduced after 2–3 hours of adenine withdrawal (0 mg/l) or limitation (2 mg/l), and was further reduced during continued growth without adenine. Contrarily, the signal of GFP lacking PDT significantly increased in intensity under the same conditions, despite the presence of mf-Lon (Fig. 3b, c). Likely, this increase reflects the reorganization and enlargement of the mito-chondrial network and volume, a characteristic of post-diauxic cells[20,22]. Together, this establishes that mf-Lon-induced mito-GFP-PDT degradation can be triggered by adenine depletion, thereby providing an inducible system for mitochondria-specific protein degradation.

### Optimizing mf-Lon expression for a minimal impact on mtDNA and nucleoid structure

High level of the endogenous Lon protease Pim1 has been shown to downregulate mitochondrial DNA content[26]. As we used the strong *ADH1* promoter ($P_{ADH1}$) for the expression of our constructs (Fig. 1a), we next performed Sybr Green I staining to analyse the number, morphology and overall distribution of mtDNA nucleoids (Supple-mentary Fig. 3a)[27]. We found that mf-Lon expression from the *ADH1* promoter left total staining intensity per cell unchanged, implying that total mtDNA content was unaltered as compared to wild type (Sup-plementary Fig. 3b). This was corroborated by mtDNA copy number measurement by quantitative PCR (qPCR) after induction of PDT degradation (Supplementary Fig. 3g). However, the total number of detected mtDNA spots per cell, representing the number of nucleoids, was decreased (Supplementary Fig. 3c). Moreover, total area occupied by mtDNA was decreased (Supplementary Fig. 3d), and the intensity and area of each nucleoid increased, as compared to wild type cells (Supplementary Fig. 3e, f). Together this shows that mf-Lon expression from the *ADH1* promoter does not affect the level of mtDNA content, but reduces nucleoid number and changes their morphology, possibly increasing their level of compaction.

To circumvent the effect of high mf-Lon expression, we sought to improve the system, aiming for a level of Lon protease closer to that of wild type. After establishing that mf-Lon expressed from $P_{ADH1}$ could perform the respiratory function of Pim1[28] (Supplementary Fig. 4a) and that Pim1 did not contribute to the GFP-PDT degradation (Supple-mentary Fig. 4c), we reduced the level of mf-Lon by replacing the *PIM1* ORF (open reading frame) with the coding sequence for MTS-mf-Lon (*pim1Δ::MITO-MF-LON*). Similar to mf-Lon expressed from *ADH1* pro-moter, mf-Lon expressed from the *PIM1* locus rescued the respiratory defect of *pim1Δ* cells (Supplementary Fig. 4a, b). Importantly, mito-chondrial GFP-PDT was also degraded in *PIM1*-replaced-mf-Lon expressing cells, albeit at a somewhat lower efficiency, in line with reduced mf-Lon expression (Supplementary Fig. 4d). Nucleoid staining also showed that even if total mtDNA content increased in the *PIM1*-replaced mf-Lon expressing cells (Supplementary Fig. 3b, g), the number of nucleoids and total area occupied by mtDNA remained at wild type levels (Supplementary Fig. 3c, d). Moreover, even though individual nucleoids stained more intensively in *PIM1*-replaced mf-Lon cells, they were significantly weaker than in cells that expressed mf-Lon from $P_{ADH1}$ promoter (Supplementary Fig. 3e, f). These data indicate that mf-Lon expression from the weaker *PIM1* promoter only has a mild effect on mitochondrial nucleoid number and morphology as com-pared to $P_{ADH1}$-controlled expression. Supporting this, *PIM1*-replaced mf-Lon cells grew like wild type cells on respiratory media, contrary to the cells expressing $P_{ADH1}$-mf-Lon, which displayed a mild growth defect (Supplementary Fig. 3h). Despite this slight inhibition of respiratory growth, the steady-state levels of the mitochondrial Hsp60 protein, which is upregulated upon mitochondrial unfolded protein stress[29], remained unaltered in both $P_{ADH1}$-mito-mf-Lon and *pim1Δ::-mito-mf-Lon* expressing strains (Supplementary Fig. 3i), indicating that mitochondrial function is largely unperturbed under conditions inducing PDT degradation.

In conclusion, lowering the level of expression of the mf-Lon protease alleviates the effects of its overexpression on mitochondrial DNA and nucleoid morphology.

### Mitochondrially targeted mf-Lon degrades endogenous PDT-tagged mitochondrial proteins

Next, we tested whether mf-Lon degrades endogenous PDT-tagged mitochondrial proteins. We analyzed two proteins which localize both to the nucleus and mitochondria: the abasic endonuclease Apn1[30], and the helicase Pif1[31], and a protein that functions exclusively inside mitochondria: the mitochondrial DNA polymerase gamma, Mip1[32]. We PDT-tagged Apn1, Pif1 and Mip1 in cells expressing mf-Lon either from

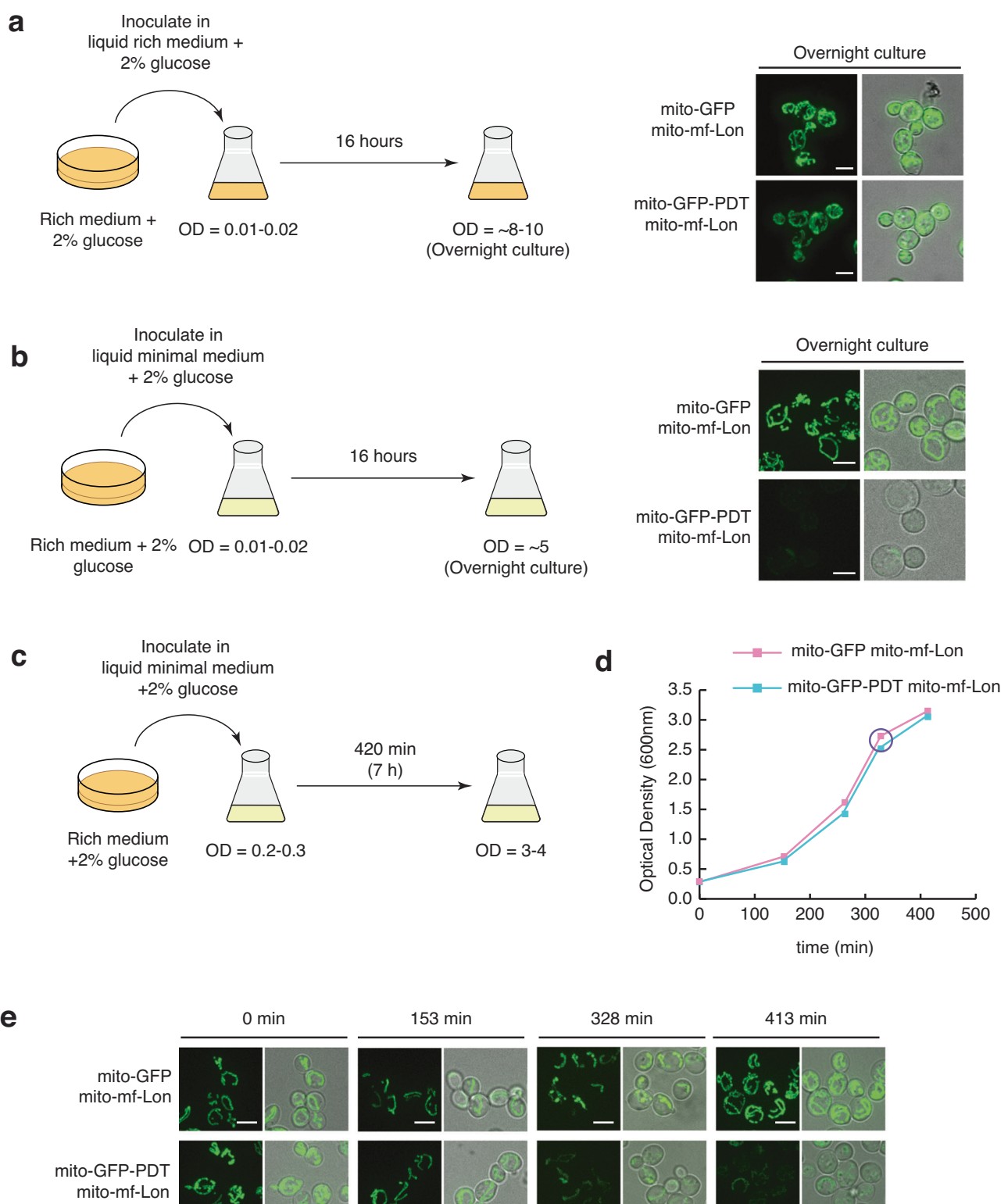

**Fig. 2 | mf-Lon-induced PDT-degradation is triggered at diauxie in minimal medium. a** Cells initiated on solid rich dextrose medium were inoculated in rich dextrose medium and grown for 16 hours, experimental schematics (left) and representative confocal microscopy images (right). Scale bar = 5 μm. **b** Similar to **a**, but initiated cultures were grown in liquid minimal dextrose medium for 16 hours before being imaged. Images in **a** and **b** are representative of results from at least three independent experiments. Cells growing in solid rich medium were inoculated in minimal medium at indicated optical density (OD) (schematics in **c**) and analyzed for growth (shown in **d**) and for GFP-PDT degradation (shown in **e**). **d** Representative growth curve of cells grown as indicated in **c**. Purple circle indicates the time-point when GFP-PDT degradation was detected. **e** Live cell confocal images of samples taken at indicated time-points in **d**. Images representative of two independent experiments. Scale bar = 5μm.

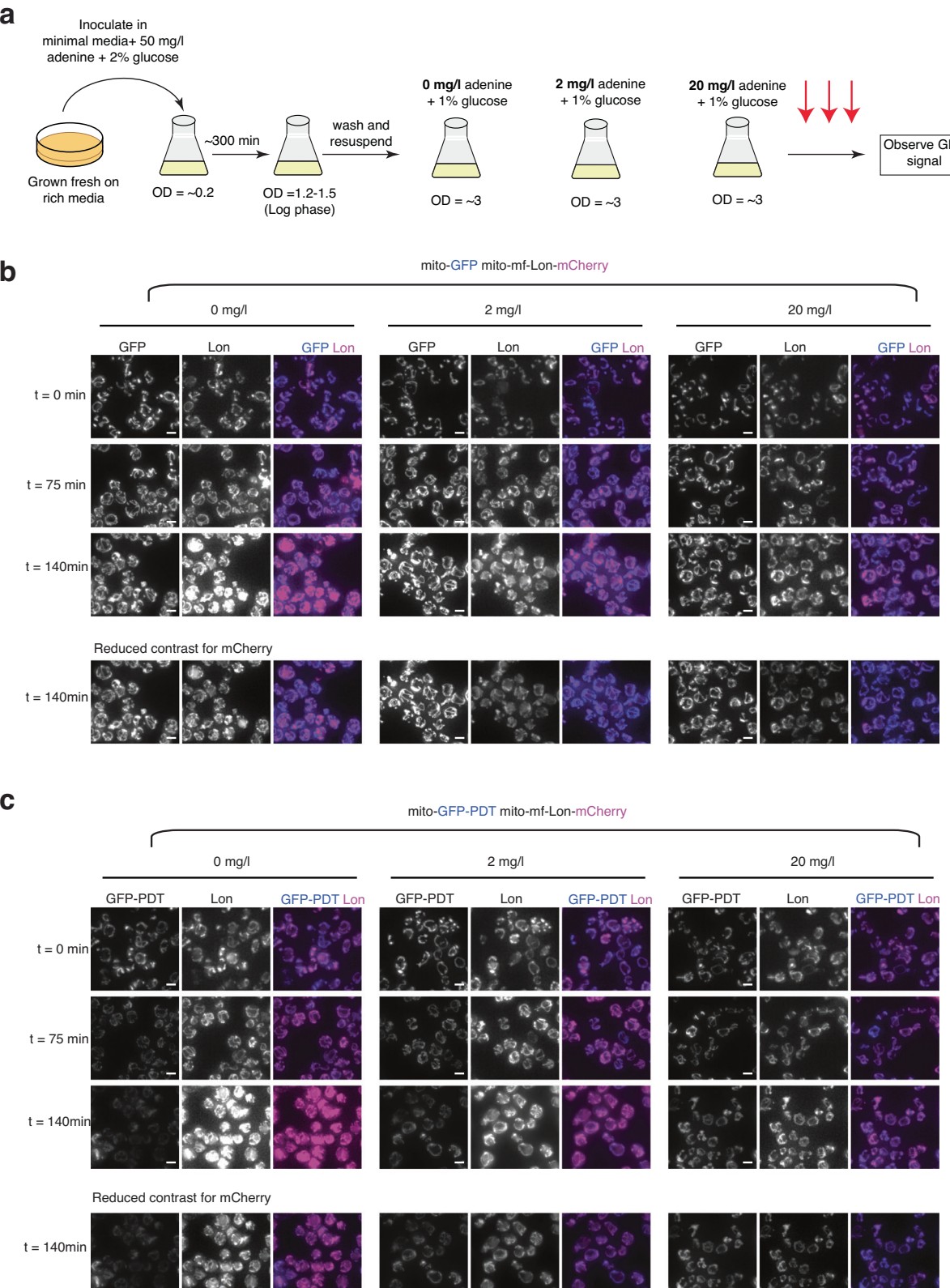

**Fig. 3 | Adenine limitation induces PDT degradation by mf-Lon protease.**
**a** Schematic for adenine titration experiment. Cells expressing mitochondrially targeted GFP and mf-Lon-mCherry, or GFP-PDT and mf-Lon-mCherry growing in 50 mg/l adenine and 2% glucose (un-inducing media) were harvested, washed and resuspended in minimal medium supplemented with 1% glucose and 0, 2, or 20 mg/l adenine. Samples were collected and examined for degradation of GFP-PDT by widefield microscopy (red arrows). **b** Live cell images of mito-GFP mito-mf-Lon-mCherry cells growing in media containing 1% glucose and 0, 2, or 20 mg/l adenine at indicated time points. **c** Same as **b** but with mito-GFP-PDT mito-mf-Lon-mCherry cells. **(b**, **c**, **bottom panels)** Reduced contrast images of indicated time-point to diminish a background signal in the channel for mCherry recording, caused by a red pigment (oxidized ribosylaminoimidazole) accumulating in the vacuole upon adenine depletion. Scale bar = 5 μm. Images are representative of three independent experiments.

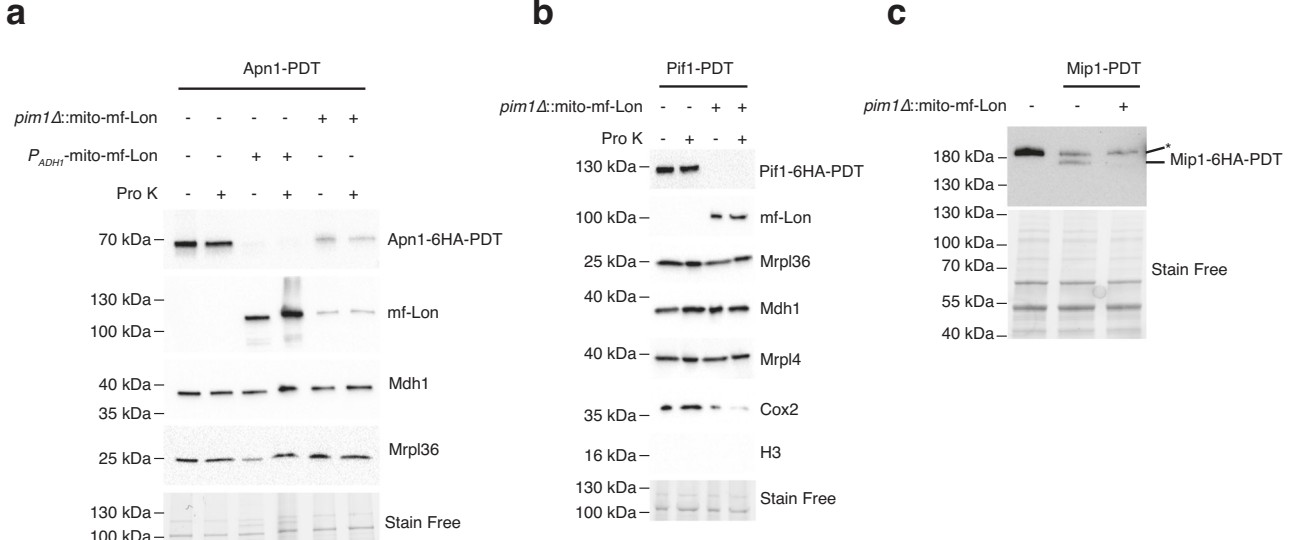

**Fig. 4 | mf-Lon protease degrades yeast endogenous mitochondrial proteins tagged with PDT. a** Western blot analysis of the mitochondria isolated from Apn1-PDT cells expressing mf-Lon from *ADH1* promoter ($P_{ADH1}$-mito-mf-Lon) or the PIM1 locus (*pim1Δ*::mito-mf-Lon) after induction of PDT-degradation. Samples were treated with Proteinase K to remove extra-mitochondrial contamination. See Supplementary Fig. 5d for additional mitochondrial markers. **b** Western blot analysis of mitochondria isolated from Pif1-PDT *pim1Δ*::mito-mf-Lon cells after induction of PDT-degradation, with or without Proteinase K treatment as indicated.

**c** Western blot analysis of whole cell extracts obtained from Mip1-PDT *pim1Δ*::mito-mf-Lon cells that were grown in minimal medium for -16 h. Data are representative of at least two independent experiments. The asterisk depicts a non-specific band. Stain-free image serves as a loading control. The sizes of the proteins are: Apn1-PDT (anti-HA), -60 kDa; Pif1-PDT (anti-HA), -100 kDa; Mip1-PDT (anti-HA), -150 kDa; H3 (anti-H3), -15 kDa; Cox2 (anti-MTCO2), -35 kDa, Mrpl4 (anti-Mrpl4), -44 kDa; mf-Lon (anti-Flag), -100 kDa; Mdh1 (anti-Mdh1), 35.6 kDa; and Mrpl36 (anti-Mrpl36), -20 kDa.

the *ADH1* promoter ($P_{ADH1}$-mito-mf-Lon) or the *PIM1* locus (*pim1Δ*::mito-mf-Lon), and grew them under the conditions that induce mito-GFP-PDT degradation (Supplementary Fig. 5a, b, Fig. 2b).

Western blot analysis of whole cell extracts revealed that the presence of $P_{ADH1}$-mf-Lon did not alter the total (nuclear and mitochondrial fractions) levels of Apn1-PDT in uninduced or induced cells (Supplementary Fig. 5c). In contrast, mf-Lon expression significantly reduced Apn1-PDT levels in crude preparations of mitochondria from induced cells (Fig. 4a, Supplementary Fig. 5c, d). This degradation was even more evident after additional proteinase K treatment, which removes unimported extra-mitochondrial proteins (Supplementary Fig. 5c). Analysis of the *PIM1*-replaced mf-Lon strain also revealed mitochondria-specific degradation of Apn1-PDT, but at lower levels, in line with the reduction of promoter-strength (Fig. 4a, Supplementary Fig. 5d). Similarly, mf-Lon expression from *PIM1* locus was sufficient for complete degradation of the lesser expressed Pif1-PDT as well as the mitochondrial Mip1-PDT (Fig. 4b, c). Importantly, except for a reduction of the respiratory Complex IV subunit, Cox2, a wide variety of mitochondrial matrix proteins, including mitochondrial ribosomes, were unaffected by mf-Lon expression (Fig. 4b, Supplementary Fig. 5c-d).

We next explored the phenotypic consequences of degrading endogenous proteins by mf-Lon on mtDNA copy number, focusing on the mf-Lon expressing from the *PIM1* locus, as this background was more similar to wild type compared to $P_{ADH1}$-mito-mf-Lon strain. We first focused on the effect of degradation of Mip1, the absence of which leads to a complete loss of the mitochondrial genome[33]. Based on this, we predicted that the degradation of Mip1-PDT by mf-Lon would result in a decline in mtDNA levels. Indeed, our qPCR results confirmed that the mf-Lon-led degradation of Mip1-PDT resulted in a complete loss of the mitochondrial genome, phenocopying *mip1Δ* mutation (Supplementary Fig. 6a). This loss of mtDNA was further underscored by the inability of these cells to grow on respiratory medium (Supplementary Fig. 6b). Thereafter, we investigated the effects of PDT degradation of the dually localized Pif1 and Apn1, and found no significant reduction

in mtDNA copy number, or ability to grow on respiratory medium (Supplementary Fig. 6a, b). This is in line with the reported subtle impact of *apn1Δ* and *pif1Δ* deletions on mtDNA copy number, that only becomes evident under conditions of genotoxic stress[30,34,35], and shows that neither of the proteins are needed for mitochondrial function during non-stressed conditions.

In summary, our findings demonstrate that mf-Lon-induced degradation of endogenous yeast mitochondrial proteins is an efficient tool for the study of mitochondrial function.

### mf-Lon-dependent mitochondrial protein degradation in human cells

Finally, as proof of concept for the versatility of our developed mitochondria-specific protein degradation tool, we investigated whether mf-Lon can induce PDT degradation in human mitochondria. To test this, we targeted mCherry-labelled mf-Lon to mitochondria and investigated if it causes PDT-tag specific degradation of a blue fluorescent reporter protein (BFP-PDT), within the mitochondria. Indeed, upon transiently expressing mitochondrially targeted mCherry-mf-Lon and either BFP, or BFP-PDT in U2OS cells, we observed a notable -30% reduction in the intensity of BFP-PDT within the mitochondria, as compared to the untagged BFP signal (Fig. 5a, b). Furthermore, when we reduced the concentration of the BFP- or BFP-PDT-expressing constructs to half, -60% of the BFP-PDT signal was lost as compared to the BFP in mf-Lon expressing cells (Fig. 5c-e), illustrating that PDT degradation by mf-Lon is dose-dependent, consistent with our observation in yeast.

In conclusion, this demonstrates that the *Mesoplasma florum* Lon protease and its corresponding degradation tag can be used for exclusive degradation of mitochondria-specific proteins in human cells.

## Discussion

The development of inducible eukaryotic degron systems has greatly enhanced our ability to investigate and understand the roles

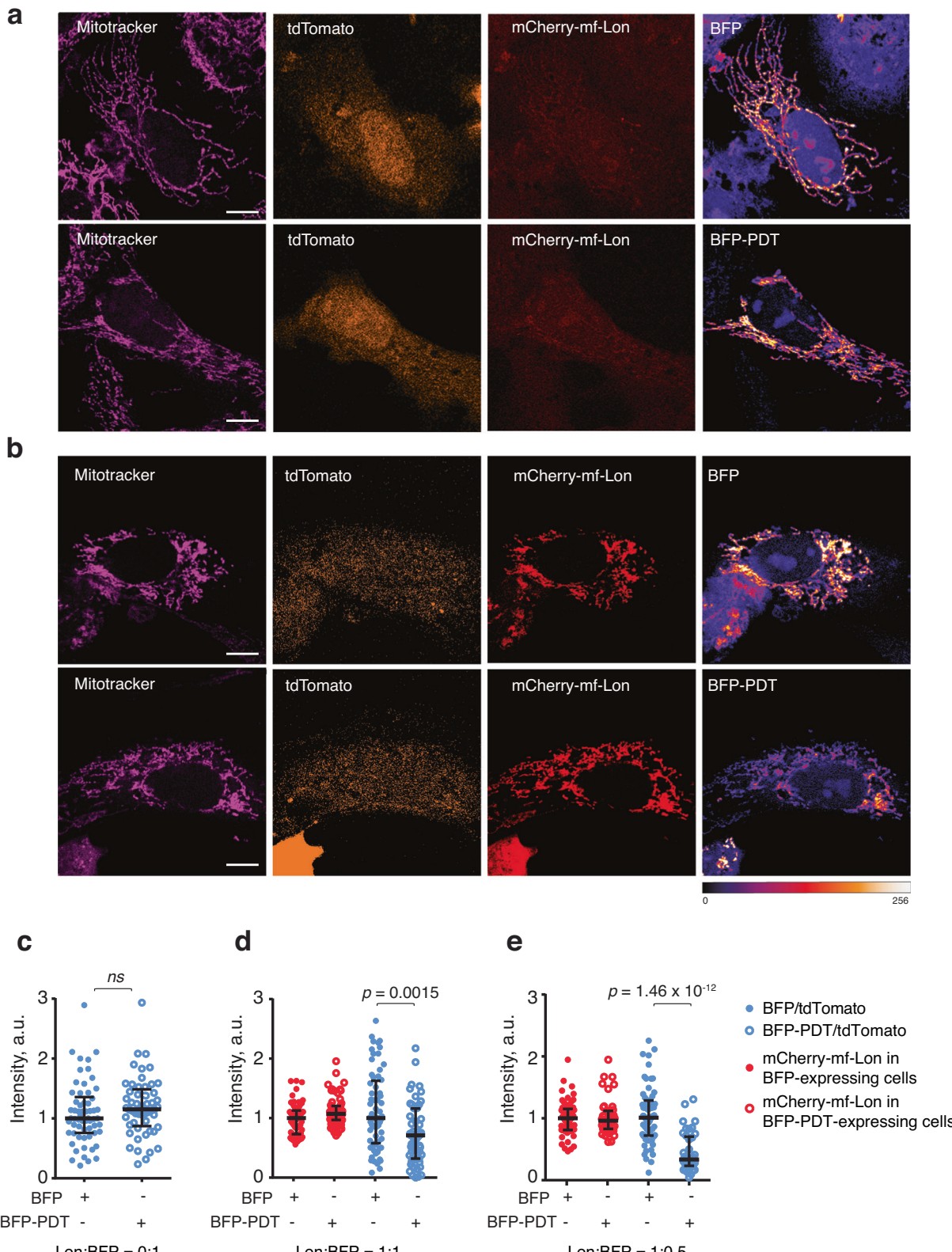

of specific proteins in various cellular functions. However, until now, a mitochondria-specific degradation system has been lacking and therefore controlled regulation of levels of mitochondrial proteins has not been possible. In our current study, we introduce a novel protein degradation system that not only targets mitochondrial proteins specifically, but also offers conditional control.

While our primary aim was to explore the functions of dually localized proteins, the potential applications of this system are broad. It can be employed for any nuclear-encoded protein involved in mitochondrial functions, circumventing the challenge of depleting proteins with functions essential for cell survival. This can provide useful insights into how these proteins contribute to mitochondrial functions and overall cellular health.

**Fig. 5 | Mitochondrially targeted mf-Lon protease specifically degrades mitochondrial BFP-PDT in human cells.** Representative images of U2OS cells 24 h after (transient) transfection with plasmids expressing mitochondrially localized BFP or BFP-PDT individually (**a**), or together with mitochondrially targeted mCherry-mf-Lon (**b**). The mitochondrial network is depicted by Mitotracker, tdTomato serves as an expression control for BFP and BFP-PDT transfections; and Fire Lookup Table (LUT) was used to visualize the BFP intensity as indicated by the colour bar (bottom). Scale Bar 10 μm. **c**–**e** Quantification of the mitochondrial BFP, BFP-PDT and mCherry-mf-Lon intensity in transfection experiments, see the detailed description in Material and Methods. Images are representative of three independent experiments. **c** Quantification of mitochondrial BFP and BFP-PDT signals for experiments shown in **a**. Each blue dot represents the ratio of intensity of BFP to tdTomato in a cell, and the data show the median value with interquartile range. BFP-PDT/tdTomato ratios were normalized to the median value of BFP/tdTomato ratio facilitating comparison between the two conditions. **d** Same as **c** except that data were plotted from experiment shown in **b**, and the intensities of mCherry-mf-Lon (red dots) were also measured and normalized to the median value of mCherry-mf-Lon intensity in BFP transfections. **e** Same as **d** except that the concentration of the BFP, or BFP-PDT plasmids were reduced by half in each transfection. The ratio of mCherry-mf-Lon plasmid concentration to BFP (or BFP-PDT) plasmid concentration used for transfection in each experimental condition is shown at the bottom in **c**–**e**. Signal intensities depicted as arbitrary units (a.u.). Pairwise comparisons were performed using 2-way ANOVA. At least 40 cells from each sample/condition were examined over three independent experiments in **c**–**e**.

We harnessed the *Mesoplasma florum* ssrA-Lon degradation process to establish our mitochondria-specific protein degradation method. The mitochondrial specificity of mf-Lon-dependent PDT degradation can be demonstrated by our cytoplasmically-localized GFP-PDT, which showed that the mf-Lon caused PDT degradation within mitochondria, but not outside (Fig. 1). This is further corroborated by the observation that mf-Lon did not significantly reduce Apn1-PDT levels in whole cell extracts but completely depleted it within the mitochondrial fraction (Supplementary Fig. 5c). This indicates that extra-mitochondrial Apn1-PDT remains unaffected and does not undergo PDT degradation outside of the mitochondria.

Lon proteases play a crucial role in protein quality control across all domains of life. The evolutionary conservation is demonstrated by the observation that the mf-Lon protease can replace yeast mitochondrial Lon (Supplementary Fig. 4a), suggesting it can also degrade bona fide Pim1 substrates, and expression of mf-Lon could impact other processes than PDT degradation. However, given that the *PIM1*-replaced-mf-Lon cells display wild-type growth on respiratory media and show no reduction in mtDNA copy number (Supplementary Figs. 3 and 6), it should be an appropriate strain for investigations using mitochondria-specific PDT degradation.

In yeast, we observed that mf-Lon-induced degradation relied on adenine auxotrophy and occurred during the diauxic shift in minimal medium, coinciding with adenine depletion[25]. Although this allowed us to induce degradation and gain more control over the system, the exact reason for this phenomenon remains elusive. Yeast strains harboring the *ade2* mutation are unable to synthesize P-ribosylaminoimidazolecarboxylate, which is necessary for adenine de novo synthesis, and they accumulate the oxidized form of ribosylaminoimidazole (resulting in a red pigment) when adenine is scarce. Intriguingly, this pigment accumulation is linked to mitochondrial function, and *ade2* mutants have been widely used to assess respiratory function through red/white colony screening[36,37]. Despite these observations, the molecular mechanism that links adenine with mitochondrial function remains unknown. One tempting hypothesis would be that adenine depletion creates a mitochondrial environment conducive to mf-Lon-mediated PDT degradation. This is also supported by our observation that removing adenine from the growth medium containing excess adenine leads to a more extended mitochondrial network (Fig. 3, Supplementary Fig. 5b), potentially indicating improved mitochondrial function. Such an environment may also be present in human mitochondria, facilitating mf-Lon-dependent PDT degradation as shown to be possible in this study. Unraveling the molecular details of the relationship between adenine (or other metabolites) and mitochondrial mf-Lon-induced PDT degradation will be an interesting study for the future.

Our experiments on mf-Lon-based targeted mitochondrial protein degradation within human cells demonstrate its evolutionary conservation and adaptability. Although further validation is needed, this represents a promising first step towards selective mitochondrial protein degradation in human cells. This is especially important given several reports suggesting the involvement of nuclear factors in the mitochondrial DNA metabolism[38–40], whose direct role within mitochondria has been questioned, as they affect the expression of mitochondrial genes in the nucleus[41]. The method presented here will be crucial to resolve this debate.

In summary, our in vivo approach for targeted mitochondrial protein degradation, significantly expands the molecular toolbox for analyzing the functions of nuclear-encoded mitoproteins within the mitochondria, and is only limited by the effects of PDT tagging of endogenous proteins. The conditional degradation of a protein of interest in yeast mitochondria is particularly powerful for investigating the roles of proteins with essential functions. Furthermore, re-introducing adenine to cultures previously induced for PDT-degradation of a protein of interest should restore its levels, providing additional insights into its molecular function. Finally, exploring different metabolism-specific promoters could potentially render this system constitutive, which in turn can provide new insights on mitochondrial dysfunction which is fundamental to neurodegenerative disorders and ageing.

## Methods
### Construction of plasmids
For mitochondrially targeted GFP, we used the pVT100U-mtGFP (Addgene) as a template for subcloning to create yeast integrative plasmids (see below).

The sequence of the Lon gene from *Mesoplasma florum* was based on the mf-Lon protein sequence obtained from NCBI database (GenBank: KM521209.1, Uniprot ID:Q6F160). A Su-9 mitochondrial targeting signal (amino acids 1-69 of subunit 9 of the F0 ATPase of *Neurospora crassa*, including a mitochondrial processing peptidase site) as described in ref. 42, and a 6xHis-3xFLAG tag were added to the N- and C-termini, respectively.

A Protein Degradation Tag (PDT) (AANKNEENTNEVPTFMLNA GQANRRRV[19]) was added to the C-terminus of GFP, and a Su-9 MTS at the N-terminus. Both constructs were codon optimized for expression in *Saccharomyces cerevisiae* (*S. cerevisiae*) and synthesized by the GeneART gene synthesis service at Thermo Fisher Scientific.

To construct yeast integrative plasmids, the MTS-mf-Lon and MTS-GFP-PDT constructs were first cloned in pVT100U-mtGFP between the *HindIII-NotI* restrictions sites, resulting in pVT100U-mtGFP-PDT and pVT100U-mt-mf-Lon plasmids. These were then digested by *SphI* to release the $P_{ADH1}$-MTS-GFP-$T_{ADH1}$, $P_{ADH1}$-MTS-GFP-PDT-$T_{ADH1}$, and $P_{ADH1}$-MTS-MF-LON-$T_{ADH1}$ fragment, respectively, and subsequently cloned into the *SphI* site of YIplac204 (GFP and GFP-PDT constructs), and YIplac128 (mf-Lon construct) integrative vectors[43], resulting in CD424, CD425, and CD429 plasmids, respectively.

The MTS was removed from YIplac204-mtGFP (CD424) and YIplac204-mtGFP-PDT (CD425) by amplifying the $P_{ADH1}$ through Polymerase Chain Reaction (PCR) and cloning the resulting DNA fragments between the *SalI-BglII* and *PciI-BglII* restriction sites of YIplac204-

mtGFP and YIplac204-mtGFP-PDT, respectively, resulting in YIplac204-GFP (CD441) and YIplac204-GFP-PDT (CD442) plasmids.

For PDT-tagging of endogenous genes, *kanMX4* from pFA6-kanMX4 was cloned into the *Not*I site of CD425 plasmid. The resulting YIplac204-mtGFP-PDT-KanMX4 (CD436) plasmid was used as a template to amplify *PDT-kanMX4* using gene-specific primers for subsequent PDT tagging using established protocols for one step epitope tagging[44] (see Supplementary Table 2).

For visualization of PDT degradation in human cells, we employed a four-colour visual system based on the pcDNA3.1(+) IRES GFP expression vector, wherein we created an MTS-mCherry-mf-Lon-IRES-GFP construct, harboring a Cox8-Su9 MTS preceding the mCherry-tagged mf-Lon, followed by an IRES (Internal Ribosome Entry Site) sequence and GFP. Similarly, we constructed an MTS-BFP (or BFP-PDT)-IRES-tdTomato expression vector. The IRES allows the co-expression of GFP and tdTomato in the cytoplasm, which serves as an internal control for the expression of MTS-mCherry-mf-Lon and MTS-BFP (or BFP-PDT), respectively.

To construct these plasmids, a sequence for Su9/Cox8 (C8S9) MTS[45] was added at the N-terminus of TagBFP, TagBFP-PDT and mCherry-mf-Lon sequences. A Kozak sequence was added at the start site of all constructs, codon optimized for expression in human cells, and synthesized by GeneArt gene synthesis service at Thermo Fisher Scientific.

The resulting C8S9-mCherry-mf-Lon, C8S9-TagBFP and C8S9-TagBFP-PDT genes were then cloned between *Nhe*I-*Not*I sites of pcDNA3.1(+)IRES GFP (Addgene plasmid #51406), resulting in pcDNA3.1-C8S9-mCh-mf-Lon-IRES-GFP (CD522), pcDNA3.1-C8S9-TagBFP-IRES-GFP, and pcDNA3.1-C8S9-TagBFP-PDT-IRES-GFP plasmids. Next, the GFP from the TagBFP plasmids was replaced by tdTomato through insertion between the *Age*I-*Pac*I sites of pcDNA3.1-C8S9-TagBFP-IRES-GFP and pcDNA3.1-C8S9-TagBFP-PDT-IRES-GFP plasmids, creating pcDNA3.1-C8S9-TagBFP-IRES-tdTomato (CD523), and pcDNA3.1-C8S9-TagBFP-PDT-IRES-tdTomato (CD524) plasmids.

CD522-524 were then used for transfection of U2OS cells. U2OS cells (ATCC version catalogue number HTB-96) resuspended in DMEM supplemented with 10% FBS were seeded on coverslips at $0.5–2 \times 10^5$ cells per well in a 24-well plate. Cells were incubated at 37 °C, 5% CO2 for 18-24 hours reaching 70-90% confluency at the time of transfection. Transfection was performed using Lipofectamine 2000 (Invitrogen) diluted in Opti-MEM (Gibco) according to the manufacturer´s instruction. 2 µl of lipofectamine and 0.8 µg of plasmid DNA were used per well for all single transfections. In co-transfection experiments, 0.8 µg of CD522 was combined with 4 µl of lipofectamine and 0.8 µg of either CD523 or CD524. To demonstrate the dose-dependency of mf-Lon-mediated mitochondrial BFP-PDT degradation, 0.8 µg of CD522 was combined with 3 µl of lipofectamine and either 0.4 µg of CD523 or CD524. Transfected cells were incubated at 37 °C, 5% CO2 for 24 hours before fixation.

## Strains
All yeast strains used are derived from CB67 (*W303, MATa, ade2-1, trp1-1, can1-100, leu2-3,112, his3-11,15, ura3, GAL, psi+, RAD5*) and listed in Supplementary Table 1. Standard transformation protocols were used. Each experiment was performed with at least two independent clones.

To obtain mitochondrial GFP visualization strains, wild-type yeast cells (CB67) were transformed with the *Bsu36*I-linearized CD424 or CD425 for integration at the *TRP1* locus. The resultant mito-GFP and mito-GFP-PDT strains were then transformed with *Afl*II-linearized CD429 plasmid for integration at the *LEU2* locus. The same procedure was used to obtain strains for cytoplasmic GFP visualization except that cells were initially transformed with *Bsu36*I-linearized CD441 and CD442, i.e., constructs lacking MTS.

*MF-LON* was tagged C-terminally with mCherry using pFA6a-link-yomCherry-Kan (Addgene plasmid #44903) as template as described in ref. 46.

The *PIM1* gene was deleted in wild type or $P_{ADH1}$-mito-mf-Lon strain by one-step replacement of *PIM1* open reading frame (ORF) with *pim1Δ::natMX4*-cassette amplified from pCloneNat1 plasmid using primers flanking the *PIM1* ORF (Supplementary Table 2). Strains containing the construct were selected on YPD agar plates supplemented with 100 mg/l of nourseothricin (Jena Bioscience, cat no. AB-101-10 ML). The *pim1Δ*::mito-mf-Lon strains were created similarly except that an *MTS-MF-LON-LEU2* DNA fragment, amplified from the YIplac204-mt-mf-Lon plasmid, was used to replace *PIM1* ORF. The correct transformants were selected on solid minimal medium lacking leucine. All integrations were confirmed by PCR.

Pif1, Apn1 and Mip1 were tagged with PDT at their C-terminus in wild-type or mito-mf-Lon strains in two steps. First, the sequence encoding a 6xHA tag was introduced at the 3'ends of endogenous *PIF1*, *APN1*, and *MIP1* genes using *HIS3*-marked pYM15 plasmid as a template for the transformation construct. Second, the 6xHA tagged genes were PDT-tagged using the *PDT-kanMX4* cassette amplified from CD436, replacing the *HIS3* marker with *kanMX4*. The correct strains were selected on YPD agar plates supplemented with 200 mg/l of G418 (Thermo Fisher Scientific, cat no 10131019), and confirmed by PCR.

## Media
Rich medium (Yeast Peptone Dextrose, YPD) consisted of 1% yeast extract (Thermo Fisher Scientific, product number 212750), 2% peptone (Thermo Fisher Scientific, product number 211677) and 50 mg/l adenine (Merck, product number A9126-100G), and was supplemented with appropriate carbon source [2% glucose (non-respiratory medium) or 3% glycerol (respiratory medium)], as indicated. Standard minimal medium (SMM) consisted of 0.67% yeast nitrogen base without amino acids (Merck, product number Y0626), supplemented with all standard amino acids [bought from Sigma (now Merck)] at 76 mg/l except leucine, which was added to 380 mg/l; adenine to 19 mg/l, and PABA to 7.6 mg/l, and 2% glucose was added as a carbon source[47].

For induction of PDT degradation, the composition of SMM was the same as described above, with the exception of adenine and glucose, which varied depending on the condition – "un-inducing" medium contained 50 mg/l adenine and 2% glucose, while "inducing" medium contained 2 mg/l adenine and 1% glucose.

For experiments when mitochondrial DNA (mtDNA) was visualized by Sybr Green I staining, SMM was supplemented with 340 mg/l isoleucine, 550 mg/l of leucine, and 430 mg/l of valine, to prevent parsing of nucleoids[48].

## Growth conditions
All experiments were initiated from fresh colonies obtained from glycerol stocks stored at −80 °C and grown on solid YPD media overnight at 30 °C. Liquid cultures were grown at 30 °C and 180 rotations per min (rpm). Each experiment was repeated at least twice.

For GFP visualization, fresh colonies were inoculated in 5 ml in SMM, and grown in 50 ml Falcon tubes for 16–18 hours, and samples were collected for GFP visualization at indicated timepoints (see Microscopy).

For growth-rate analysis, cells were inoculated in SMM at indicated optical densities ($OD_{600}$), and growth was estimated by $OD_{600}$ measurements at indicated timepoints.

## Induction of PDT degradation
For adenine titrations (Fig. 3), mito-GFP + mito-mf-Lon-mCherry and mito-GFP-PDT + mito-mf-Lon-mCherry cells were grown to logarithmic growth (log phase, $OD_{600} = 1.2–1.5$), after which cells were pelleted, washed with 1x PBS buffer, and resuspended in SMM containing 1% glucose and 0 mg/l, 2 mg/l, or 20 mg/l adenine at a cell concentration of $OD_{600} = ~3$, and thereafter grown for 3 hours at 30 °C and 180 rpm. Samples were collected at indicated timepoints and imaged as described below (see Microscopy).

For induction of Apn1-PDT and Pif1-PDT degradation, cells were inoculated in 100 ml SMM supplemented with 50 mg/l adenine and 2% glucose at OD = 0.2 and grown for 6 hours to obtain a log phase culture ($OD_{600}$ = 1). The cultures were then expanded by re-inoculation in 400 ml of fresh, un-inducing medium and grown overnight to an $OD_{600}$ of 7–8. Cells were subsequently pelleted by centrifugation (4000 rcf, 5 min, room temperature), washed once with sterile water, and resuspended in 400 ml of inducing medium (SMM supplemented with 2 mg/l adenine and 1% glucose). To ensure that PDT degradation was induced, mito-GFP + mito-mf-Lon and mito-GFP-PDT + mito-mf-Lon cells were grown in parallel with the experimental strains, and PDT degradation was analyzed by microscopy. Furthermore, the accumulation of red pigment resulting from adenine depletion served as a visual reference of diauxie[25,49], which usually occurred 3–4 h after the shift to the inducing medium.

For induction in wild type, $P_{ADH1}$-mito-mf-Lon, and $pim1\Delta$::mito-mf-Lon strains, cells were first grown in 20 ml un-inducing medium to reach log phase ($OD_{600}$ = 1). Subsequently, cells were reinoculated in 50 ml un-inducing medium and grown overnight to an $OD_{600}$ = 8–10 (30 °C, 180 rpm, doubling time = 1.667 h). Finally, cells were washed and shifted to 50 ml of inducing medium and grown for ~4 h. Samples were collected before and after induction for estimation of mtDNA by quantitative real time PCR (qPCR).

### Estimation of mtDNA copy number by qPCR
Wild type, $P_{ADH1}$-mito-mf-Lon, $pim1\Delta$::mito-mf-Lon strains were induced for PDT degradation, and 2 ml of culture was harvested before and after shifting cells to the inducing media. Total DNA was isolated by phenol:chloroform:isoamyl alcohol and cold ethanol precipitation method. Relative quantity of mtDNA was estimated by qPCR using primers against *COX2* and *ACT1* (Supplementary Table 2) and a standard curve generated from total DNA isolated from an uninduced culture of wild type with Fast SYBR Green (Applied Biosystems, 4385612) according to manufacturer's protocol. Unpaired, 2-tailed *t*-test was performed to analyze data with error bars representing standard deviations from 3 independent experiments. The same procedure was applied for measurement of mtDNA copy number in strains shown in Supplementary Fig. 6a, except that the cells were grown overnight (~16 h) in 5 ml standard minimal medium to $OD_{600}$ of ~5-6. CFX Maestro Software (Bio-Rad) Version 2.3 was used for collection and analysis of the data.

### Isolation of mitochondria
Mitochondria were prepared from yeast cells by differential centrifugation method as described in[50]. Briefly, 400 ml of Apn1-PDT and Pif1-PDT cells, expressing mf-Lon from the *ADH1* promoter, or from *PIM1* locus, induced for PDT degradation, were harvested (5000 *g*, 10 min, room temperature), resuspended in 2 ml of MP1 buffer (100 mM Tris, 10 mM DTT) for every gram of cell pellet, and incubated (30 °C, 10 min). The cells were subsequently washed with 1.2 M sorbitol and resuspended in MP2 buffer [1.2 M Sorbitol, 0.02 M potassium phosphate buffer, pH 7.4, 20 T Zymolyase (Seikagaku Biobusiness, product code 120491) at 3 mg/g cell pellet] using 6.7 ml per gram of cell pellet, followed by incubation (30 °C, 60 min) to create spheroplasts which lack the yeast cell wall. At this point, a 50 µl cell extract sample was taken and boiled for 5 min with 50 µl of 5x SDS loading buffer and stored at −20 °C until analysis by Western blotting.

The remaining spheroplasts were centrifuged (4500 *g*, 6 min at 4 °C), and resuspended in MP3 buffer (10 mM Tris pH 7.4, 1 mM EDTA, 1 mM PMSF, 0.6 M sorbitol) using 13.4 ml per gram of cell pellet. This spheroplast suspension was then homogenized using a tissue grinder (Fisherbrand, product code 10331592), centrifuged (4500 *g*, 6 min at 4 °C), and the supernatant was cleared of any remaining cell debris by two additional centrifugations (4500 *g*,

6 min, at 4 °C). The resulting supernatant was re-centrifuged (20,000 *g*, 25 min at 4 °C), and the pellet, representing a crude preparation of mitochondria was resuspended in SH buffer (0.6 M sorbitol, 20 mM HEPES pH 7.4) to a concentration of 10 mg/ml. These were finally snap-frozen in liquid nitrogen and stored at −80 °C.

### Proteinase K treatment of mitochondria
600 µg of mitochondria were harvested in two 1.5 ml Eppendorf tubes (300 µg each) by spinning (10,000 *g*, 10 min at 4 °C) in a tabletop centrifuge. 40 µl of SH buffer was added to the pellet in each tube and incubated on ice for 10 min. Then, 5 µl of 0.5 mg/ml of Proteinase K (Sigma RPROTK-RO, dissolved in SH buffer at 50 µg/ml) was added to one of the tubes, and both tubes were incubated on ice for another 10 min. Thereafter, 5 µl of 100 mM PMSF was added to both tubes, that were subsequently centrifuged (21,000 *g*, 10 min, at 4 °C). The resulting mitochondrial pellets were washed twice in SHKCL buffer (0.6 M sorbitol, 20 mM HEPES pH 7.4, 150 mM KCL) by centrifugation (21,000 *g*, 10 min at 4 °C), and finally resuspended in 100 µl of 2.5x SDS loading dye. These were then boiled for 10 min, supernatant collected by centrifugation (20,000 *g*, 10 min, 4 °C) and stored at −20 °C until analysis by Western blotting.

### Western blotting
For cell extract samples from induced culture, 50 µl sphaeroplasts suspension was boiled directly with 2.5x SDS loading buffer before analysis. The cell extract protein sample from uninduced culture was prepared by the standard trichloroacetic acid precipitation (TCA) method. Briefly, 250 µl of 20% TCA was added to 10 ml of pelleted cell culture, and the mixture was vortexed in the presence of glass beads to disrupt the cells. The suspension was pelleted at 13000 rpm, 4 min at room temperature, and washed twice with ice-cold 100% ethanol and air-dried. The dried pellets were subsequently resuspended in 2.5x SDS loading buffer and boiled for 5 min. Cell debris was removed by centrifugation (20,000 *g*, 10 min, 4 °C) and the supernatant was stored at −20 °C before analysis by Western blotting according to standard protocols. The following antibodies were used: PDT-tagged proteins - anti-HA antibody (Sigma, cat# 11666606001, dilution 1:1000), mf-Lon - anti-Flag (Sigma-Aldrich, cat# F1804-5 MG, dilution 1:3000), histone - anti-H3 (Abcam, cat# ab1791, dilution 1:1000), HSP60 – anti-HSP60 monoclonal antibody LK2 (AH Diagnostics, cat# ADI-SPA-807-E, dilution 1:1000), GFP - anti-GFP (Abcam, cat# ab6556, dilution 1:1000); Cox2 – anti-MTCO2 (Abcam, cat# ab110271, dilution 1:1000); the following antibodies were gifts from Martin Ott: Mrpl4 – anti-Mrpl4 (dilution 1:2000), Mrpl36 – anti-Mrpl36 (dilution 1:500), Mrpl40 – anti-Mrpl40 (dilution 1:2000), Mdh1 – anti-Mdh1 (dilution 1:10,000). The following secondary antibodies were used: Polyclonal Goat Anti-Rabbit Immunoglonulins/HRP (DAKO, cat# P0448, dilution 1:5000); used for antibodies against Mrpl4, Mrpl36, Mrpl40, Mdh1, H3, GFP. Goat Anti-Mouse Immunoglobulins/HRP (DAKO, cat# P0447, dilution 1:5000, with exception in the bracket below); used for antibodies against HA, FLAG (1:10,000), Cox2, Hsp60.

### Visualization of mtDNA
Wild type, $P_{ADH1}$-mito-mf-Lon, $pim1\Delta$::mito-mf-Lon strains were grown overnight in SMM with excess isoleucine, leucine and valine (SMM + ILV) to $OD_{600}$ = 8-10 (see Media). The excess branched-chain amino acids did not affect mf-Lon-induced PDT degradation (data not shown). One milliliter of overnight culture was collected and washed with PBS using centrifugation (3000 rpm, 2 min), and incubated for 30 s at room temperature in 500 µl staining solution containing 2.5 µl of Sybr Green I (Thermo Fisher Scientific, S7563) in PBS. Finally, cells were washed twice with PBS using centrifugation (3000 rpm, 2 min) before imaging in SMM + ILV medium.

## Microscopy

70 µl of cells were immobilized on glass bottom wells of imaging plates (Mobitec, Imaging Plate 96 CG 1.5, Order #: 5242-20) coated with Concanavalin A (0.45 mg/ml, Sigma) and overlaid with appropriate media. For confocal microscopy (Figs. 1b, 2b, e, Supplementary Fig. 1a and 4d), cells were imaged in a full-size temperature-controlled incubator (set at 30 °C throughout the experiment) on a Nikon Ti Eclipse microscope equipped with a spinning disk confocal (Yokagawa CSU-X1), an ALC laser controller, and an Andor DU-897 X-3655 EM-CCD camera (Pixel size 16 µm. QE 95%), using a Nikon 100x/1.4 PlanApo oil objective. A 1.2x lens was inserted in the emission light path to fulfil the Nyquist sampling theorem with the camera used. The acquisition settings for GFP were 488 nm diode laser, 9.4%, 50 ms exposure time, 7.5 µm Z-stack with 0.3 µm interval, and for mtDNA staining with Sybr Green I (Supplementary Fig. 3a), 488 nm diode laser at 9.8% power, 20 ms exposure time.

For widefield microscopy (Figs. 1d, 2a, 3b, c, Supplementary Figs. 2a and 4c), cells were imaged in the same environmental conditions and on the same microscope as above, but using an Andor Zyla 4.2+ sCMOS camera (pixel size 6.45 µm, QE 82%) without the 1.2x lens, a Lumencore SpectraX LED light source, a Sutter external emission filter wheel, a 60x/1.2 PlanApo water objective, and a 1.5x magnification lens inserted, in order to fulfill Nyquist sampling theorem. The acquisition settings were the following: Lon-mCherry – excitation filter 555 nm at 84%, emission filter 630/90 nm, exposure time 400 ms; mito-GFP – excitation filter 470 nm at 22%, emission filter of 525/30 nm, exposure time 30 ms; cyto-GFP – excitation filter 470 nm at 50%, emission filter 525/30 nm, exposure time 200 ms, 5.55 µm Z-stack with 0.37 µm interval.

Transfected U2OS cells were grown on glass coverslips in 24-well plates and fixed in 3.7% Formaldehyde for 30 min at 37 °C. Prior to fixation, cells were incubated in 100 mM Mitotracker Deep Red (ThermoFisher Scientific) for 20 min to visualize the Mitochondria. Fixed cells were briefly washed in PBS and mounted in Prolong Diamond Antifade (ThermoFisher Scientific). Slides were cured for 24 hours at room temperature and single Z-plane images were captured on Zeiss LSM800, equipped with the Airyscan detector or Leica Stellaris 5 scanning confocal microscopes using 63x/1.4 Plan-Apo oil objective. All slides within the same transfection experiment were imaged with consistent settings and processed concurrently to ensure minimal variations between slides.

## Image analysis

Images of mitochondrial GFP captured by widefield microscopy were processed to remove the background using the FAST deconvolution or Denoise.ai functions of NIS-elements imaging software (Nikon Instruments, Tokyo, Japan). All images were colorized in the Fiji software and shown as maximum-intensity Z-projections. The brightness and contrast values were set equally along all samples.

For quantification of fluorescent signals, yeast cell segmentation was performed by YeastSpotter (http://yeastspotter.csb.utoronto.ca)[51]. Cell quantification was performed in Fiji[52] on maximum intensity Z-projections. Cells that had an area less than 10 µm² or circularity less than 0.85 were excluded from the analysis. For mito-GFP/GFP-PDT and cyto-GFP/GFP-PDT expressing cells, the intensity was estimated using integrated density in the whole cell after background subtraction. To minimize experiment-to-experiment variability, the values were normalized to the median level displayed by cells expressing GFP only.

For nucleoid analysis after Syber Green I staining, the spots segmentation was performed in Fiji on maximum intensity Z-projections after computing Laplacian image using FeatureJ plugin (http://imagescience.org/meijering/software/featurej/). To exclude noise/debris and occasional weakly stained nucleus DNA, spots that displayed area less than 0.06 µm² or median integrated density less than 500 a.u. (arbitrary units) were excluded from the analysis. To adjust nucleoid number in occasional aggregates, we divided each spot integrated density by the median spot integrated density found on the image and rounded the result to the nearest integer above zero. To minimize experiment-to-experiment variability, values were normalized to the average level displayed by the wild-type cells (CB67). The outliers were excluded by IQR method. Statistical analysis was performed by *Hierarch* resampling Python-based module[53], and a *P* value of less than 0.05 was considered statistically significant. Graphical data presentation was done by GraphPad Prism 9.0 (www.graphpad.com).

For analysis of mf-Lon-dependent mitochondrial BFP-PDT degradation, the cell area was manually delineated on a single Z-plane image, and an automatic thresholding method (Triangle procedure) in the Fiji software was utilized to create a mask based on the mCherry-mf-Lon channel. For slides where mCherry-mf-Lon was not transfected, the mask was generated based on the BFP channel. The cells where the masked area was below 10 µm² were excluded from analysis. The mean intensity of mCherry-mf-Lon and mito-BFP (or mito-BFP-PDT) was then determined within the mask using Fiji software, and background signal levels of the mito-BFP (or mito-BFP-PDT) were measured in regions devoid of mitochondria and subsequently subtracted. The mean intensity of tdTomato was measured within manually selected regions of interest that were free from mitochondria.

To estimate the degradation of mito-BFP-PDT in the presence of mf-Lon, the ratio of mito-BFP or mito-BFP-PDT to tdTomato was calculated. To facilitate comparisons across different experiments, the median level of mito-BFP/tdTomato in each experiment was standardized to 1. Similarly, the intensity of mCherry-mf-Lon was normalized to the median intensity of mCherry-mf-Lon in transfections involving mito-BFP. The outliers were excluded by IQR method.

Graphs were generated using Prism software, displaying pooled data from three independent experiments. Statistical analysis was carried out using a 2-way ANOVA in R version 4.1.2[54], using "car" package[55], and a *P* value of less than 0.05 was considered statistically significant.

## Reporting summary

Further information on research design is available in the Nature Portfolio Reporting Summary linked to this article.

## Data availability

All data are available in the main text, supplementary information, and/or source data. All materials used in the investigation are available upon request to corresponding authors. Databases used to retrieve mf-Lon gene and protein sequences were NCBI [https://www.ncbi.nlm.nih.gov/nuccore/KM521209] and UniProt (accession number Q6F160), respectively. *Saccharomyces* genome database was used to obtain sequence information for *PIF1*, *APN1*, and *MIP1* genes. Source data are provided with this paper.

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

## Acknowledgements

We are thankful to Maria Falkenberg, Martin Ott and the members of C.B. lab for helpful discussions. We thank Martin Ott for antibodies directed against mitochondrial matrix proteins. This study was funded by a post-doctoral grant from Wenner-Gren Foundations (UPD2018-0192) to S.S., a Swedish Research Council grant (DNR 2021-01118) to A.K., a project grant from the Swedish Cancer society to L.S. (DNR 22 2387), and a Knut and Alice Wallenberg foundation project grant to C.B. Microscopy was performed at the Live Cell Imaging Core facility/Nikon Center of Excellence, at Karolinska Institutet, supported by the KI infrastructure council.

## Author contributions

S.S. and C.B. conceptualized the project. S.S. designed and tested the mf-Lon-dependent mitochondria-specific PDT degradation system in yeast, set up the adenine-based induction of the system, and confirmed PDT degradation of endogenous proteins and performed qPCRs. A.K. performed quantifications, statistical analyses, imaging and analysis of the PDT degradation in human mitochondria. S.S., L.S. and A.K. designed the four-color system for visualization of PDT degradation in U2OS cells. L.S. created the constructs and performed transfections in U2OS cells. The project was overall administered and supervised by C.B. S.S. wrote the original draft. All authors edited and revised the manuscript.

## Funding

## Competing interests

The authors declare no competing interests.
