## [Peer Review File · Nature Communications]

A system for inducible mitochondria-specific protein degradation in vivoREVIEWER COMMENTS

Reviewer #1 (Remarks to the Author):

In this manuscript, Sanyal and colleagues present a method for inducible mitochondria-specific protein degradation in yeast cells. The approach is based in a degron-like system consisting of the *Mesoplasma florum* Lon (mf-Lon) protease and its corresponding *ssrA* tag (PDT). The manuscript presents proof-of-concept experiments in different media composition and growth phase conditions, assessing the efficacy of the system using mitochondrion-targeted fluorescent proteins. Finally, the system is tested in two dual-localized mitochondrial proteins and shows that it can be useful to study this kind of proteins.

Some points should be addressed before publication:

1. The system is shown to be inducible by adenine, however the mechanism behind this regulation is not properly addressed. If it is just before adenine depletion slows growth even in the presence of glucose, as stated in line 75, then many other metabolites should do the same trick.
2. The authors should discuss the applicability of their system to other organisms, particularly humans. As explained, it seems that the system can only be used for yeast adenine auxotrophic mutants.
3. In line 139, it is stated: "Similarly, mf-Lon expression from PIM1 locus was sufficient for complete degradation of the lesser expressed Pif1-PDT (Fig. 4b). Importantly, except for a modest reduction of the respiratory Complex IV subunit, Cox2, a wide variety of mitochondrial matrix proteins, including mitochondrial ribosomes, were unaffected by mf-Lon expression". In truth, the COX2 steady-state levels are markedly decreased. However, is not this expected, given the proposed role of mitochondrial Pif1 in mtDNA maintenance? Please clarify this point.

Reviewer #2 (Remarks to the Author):

The manuscript by Sanyal et al. aims to establish an inducible mitochondria-specific protein degradation system using a mycoplasma AAA+ Lon protease, which via a so-called *ssrA*-tag degrades in its native context nascent polypeptide from stalled ribosomes. Given that several mitochondrial proteins show dual localisation also to other cellular organelles, and mitochondria protein import follows complex mechanisms making it challenging to mutagenise some targeting signals, a system that specifically removes protein in the mitochondrial matrix may be of general relevance.

However, the manuscript does not convincingly characterise the mechanism underlying this approach. Likewise, the important control of a cytosolically localised Lon protease that is predicted not to affect mito-GFP harbouring the *ssrA*-tag but instead to target cyto-GFP-*ssrA*, is missing. Specificity should be addressed by proteomics or at least by a bioinformatic-based analysis of the predicted susceptibility of the yeast proteome. Moreover, the observation that Lon-induced degradation depends on the growth conditions (requiring adenine depletion after diauxic shift from fermentation to respiration) limits the potential of the outlined workflow. E.g. it is not conceivable that the system can be transferred to higher eukaryotes. Hence, the proof-of-concept experiment shown in Fig. 4 targeting two proteins with known

dual localisation may be extended by further analysing the specific function of the mitochondrial pool of the selected candidates in order to showcase the power of the system in combination with yeast as a model organism.

Reviewer #3 (Remarks to the Author):

Sanyal and colleagues present an interesting approach to selectively deplete a mitochondrial pool of specific proteins. Such technique is interesting when the function of the mitochondrial fraction of a protein with a dual localization is analyzed. The technique is based on the expression of a Lon protease from the bacterium *Mesoplasma florum* in baker's yeast. Mf-Lon degrades proteins with a ssrA tag, termed PDT. Depletion of adenine in the growth media induces the degradation of PDT-tagged proteins inside mitochondria by mfLon. The experimental approach is interesting to analyze the functions of dually localized proteins. There are some concerns that the authors should address.

The Lon protease degrades many mitochondrial proteins. Does the expression of an additional Lon protease affect mitochondrial proteostases and cause pleiotropic effects. The authors should show steady state levels of stress marker proteins and show a growth test of the used strains.

The authors should also show that mfLon localizes to mitochondria.

The authors should comment on why mfLon is not induced on full medium.

Figure 1B: Western blot would be important to reveal the depletion of the PDT-tagged proteins.

The authors present that mitochondrial Apn1-PDT and Pif1-PDT are degraded by the used approach. To show the selective depletion of the mitochondrial pool the authors should show that the nuclear fractions of the proteins unaffected.

The authors state that they used proteinase K to remove nuclear contamination from the mitochondrial isolation. However, proteinase K treatment is typically used to remove non-imported proteins from the mitochondrial surface, but not entire organelles. Furthermore, also Cox2 is reduced after proteinase K treatment (Figure E5c), indicating that the mitochondria are not fully intact. The authors should comment on this.

Is this approach specific for yeast cells or could it also be used in mammalian cell lines?

AUTHOR REBUTTAL

We thank the reviewers for their time and comments that have helped us to improve our manuscript. Most importantly, our new results show that the system for mitochondria-specific protein degradation also functions in human cells. This answers a major concern raised by all three reviewers and shows that the mf-LON protease works outside the metabolic shift required for its activity in yeast. More detailed information is found in our point-by-point replies below.

Reviewer #1 (Remarks to the Author):

In this manuscript, Sanyal and colleagues present a method for inducible mitochondria-specific protein degradation in yeast cells. The approach is based in a degron-like system consisting of the *Mesoplasma florum* Lon (mf-Lon) protease and its corresponding *ssrA* tag (PDT). The manuscript presents proof-of-concept experiments in different media composition and growth phase conditions, assessing the efficacy of the system using mitochondrion-targeted fluorescent proteins. Finally, the system is tested in two dual-localized mitochondrial proteins and shows that it can be useful to study this kind of proteins.

Some points should be addressed before publication:

1. The system is shown to be inducible by adenine, however the mechanism behind this regulation is not properly addressed. If it is just before adenine depletion slows growth even in the presence of glucose, as stated in line 75, then many other metabolites should do the same trick.

We indeed find that in yeast, mf-Lon activity is induced by the diauxic shift (the transition from fermentative to respiratory growth) resulting from depletion of adenine from the media. Additionally, the degradation of GFP-PDT is entirely reliant on the adenine mutation, as we did not observe PDT degradation in *ADE2*-proficient strains (data not presented). Yeast strains harbouring the *ade2* mutation are unable to synthesize P-ribosylaminoimidazolecarboxylate— a product required for adenine *de novo* synthesis – and accumulate oxidized ribosylaminoimidazole (manifested by a red pigment) when adenine is depleted¹. Intriguingly, this red pigment accumulation depends on mitochondrial function, and *ade2* mutants have been widely used in screens for respiratory function through red/white

colony screening ¹. Despite these observations, the molecular mechanism that links adenine and mitochondrial function remains unclear.

Significantly, we noticed that following adenine depletion from the culture medium, the mitochondrial network becomes more extensive, resembling the mitochondria in cells cultivated in respiratory conditions (see Figure 3 and reference ²). This could indicate that adenine depletion enhances the respiration of *ade2*-deficient yeast strains, thereby creating a mitochondrial environment that allows mf-Lon-dependent PDT degradation, and indeed could involve other metabolites that promote respiration. However, this hypothesis requires further investigation, and, especially in the light of our new results showing that mitochondria-specific mf-Lon degradation functions in human cells, we find that such analysis lies outside the scope of this investigation.

We have now included a discussion on these points in the revised manuscript (lines 246-264).

2. The authors should discuss the applicability of their system to other organisms, particularly humans. As explained, it seems that the system can only be used for yeast adenine auxotrophic mutants.

We now show that mf-Lon indeed degrades mitochondrially-targeted BFP-PDT in human cells, showing that the system has a broader applicability than yeast adenine auxotrophic mutants (Figure 5). Potentially, this indicates that adenine depletion of the yeast mutants creates a mitochondrial environment more like that present in human cells. As stated, this will be an interesting future topic of investigations.

3. In line 139, it is stated: “Similarly, mf-Lon expression from PIM1 locus was sufficient for complete degradation of the lesser expressed Pif1-PDT (Fig. 4b). Importantly, except for a modest reduction of the respiratory Complex IV subunit, Cox2, a wide variety of mitochondrial matrix proteins, including mitochondrial ribosomes, were unaffected by mf-Lon expression”. In truth, the COX2 steady-state levels are markedly decreased. However, is not this expected, given the proposed role of mitochondrial Pif1 in mtDNA maintenance? Please clarify this point.

We not only notice a reduction of Cox2 when Pif1 is degraded, but also after depletion of Apr1 (Extended Data Figure 5c-d) and the GFP reporter protein (data not shown). This indicates that mf-Lon expression in itself causes the reduction of Cox2, which in turn might

be caused by the perturbation of mitochondrial DNA morphology detected in *P_{ADHI}*-mito-mfLon and *pim1Δ*::mito-mfLon expressing cells (Extended Data Fig 3a-g). However, all other mitochondrial markers tested remain unperturbed (Fig 4a-b, Extended Data Fig 5c-d), and *pim1Δ*::mito-mfLon cells display wild type growth on respiratory medium (Extended Data Fig 3h, 6b), and only mild mitochondrial DNA perturbations. This indicates that the *pim1Δ*::mito-mfLon strain offers a more appropriate system to study mitochondria-specific protein degradation. This is presented in the Results section under the heading “**Optimizing mf-Lon expression for a minimal impact on mtDNA and nucleoid structure**” and discussed in lines 234-245.

Reviewer #2 (Remarks to the Author):

The manuscript by Sanyal et al. aims to establish an inducible mitochondria-specific protein degradation system using a mycoplasma AAA+ Lon protease, which via a so-called *ssrA*-tag degrades in its native context nascent polypeptide from stalled ribosomes. Given that several mitochondrial proteins show dual localisation also to other cellular organelles, and mitochondria protein import follows complex mechanisms making it challenging to 3utagenize some targeting signals, a system that specifically removes protein in the mitochondrial matrix may be of general relevance.

However, the manuscript does not convincingly characterise the mechanism underlying this approach. Likewise, the important control of a cytosolically localised Lon protease that is predicted not to affect mito-GFP harbouring the *ssrA*-tag but instead to target cyto-GFP-*ssrA*, is missing. Specificity should be addressed by proteomics or at least by a bioinformatic-based analysis of the predicted susceptibility of the yeast proteome. Moreover, the observation that Lon-induced degradation depends on the growth conditions (requiring adenine depletion after diauxic shift from fermentation to respiration) limits the potential of the outlined workflow. E.g. it is not conceivable that the system can be transferred to higher eukaryotes. Hence, the proof-of-concept experiment shown in Fig. 4 targeting two proteins with known dual localisation may be extended by further analysing the specific function of the mitochondrial pool of the selected candidates in order to showcase the power of the system in combination with yeast as a model organism.

To clarify the rebuttal, we have dissected the comments from the text above:

1. “However, the manuscript does not convincingly characterise the mechanism underlying this approach”

We agree that our work does not completely define the mechanism behind the observed mf-Lon-mediated degradation of PDT tagged proteins, but based on the following we believe that such detailed analysis lies outside the scope of this investigation. We show that in yeast mf-LON activity is induced by the diauxic shift (the transition from fermentative to respiratory growth) resulting from depletion of adenine. Additionally, GFP-PDT degradation is entirely reliant on the adenine mutation, as we did not observe PDT degradation in *ADE2*-proficient strains (data not presented). Yeast strains harbouring the *ade2* mutation are unable to synthesize P-ribosylaminoimidazolecarboxylate – a product required for adenine *de novo* synthesis – and accumulate oxidized ribosylaminoimidazole (manifested by red pigment) when adenine is depleted ¹. Intriguingly, this red pigment accumulation depends on mitochondrial function, and *ade2* mutants have been widely used in screens for respiratory function through red/white colony screening ¹. Despite these observations, the molecular mechanism that links adenine and mitochondrial function remains unclear.

Significantly, following adenine depletion from the culture medium, the mitochondrial network resembles mitochondria in cells cultivated in respiratory conditions (see Figure 3 and reference ²). This could indicate that adenine depletion enhances the respiration of *ade2*-deficient yeast strains, thereby creating a mitochondrial environment that allows mf-Lon-dependent PDT degradation. However, this hypothesis requires further investigation, and in the light of our new results showing that mitochondria-specific mf-Lon degradation functions in human cells, we find that such analysis lies outside the scope of this investigation.

We have now included a discussion on these points in the revised manuscript (lines 246-264).

2. “Likewise, the important control of a cytosolically localised Lon protease that is predicted not to affect mito-GFP harbouring the *ssrA*-tag but instead to target cyto-GFP-*ssrA*, is missing”.

We did not explore the degradation of cytoplasmic GFP-PDT by cytoplasmic Lon, as our

primary objective was the development of a mitochondria-specific protein degradation system. Previous studies have demonstrated the capability of the *E. coli* ClpXP proteasome to degrade ssrA-tagged GFP in the yeast cytoplasm ³, suggesting that cyto-mf-Lon will likely target cyto-GFP-PDT for degradation. However, this was not our focus, given the abundance of degron systems available for cytoplasmic and nuclear functionality.

3. “Specificity should be addressed by proteomics or at least by a bioinformatic-based analysis of the predicted susceptibility of the yeast proteome”

We have performed bioinformatics to investigate the potential presence of mf-Lon targets in the yeast proteome. An important criterion for such targets would be high sequence similarity with PDT#3 or PDT (the original mf-ssrA tag) sequence at the C terminus, as mf-Lon degrades proteins that are tagged C-terminally with ssrA ⁴.

Our approach involved BLASTp analyses using the *Saccharomyces cerevisiae* RefSeq as well as non-redundant (nr) protein databases to identify homologous sequences to PDT#3 and PDT. Given that mf-Lon performs PDT degradation exclusively inside the mitochondria (Figure 1C), we concentrated on proteins known to have mitochondrial localization. We ranked the hits according to *E*-values, which is a score inversely proportional to sequence homology, and looked for candidates with *E*-values < 0.01, an accepted threshold for sequences to be considered significantly similar ⁵.

The smallest *E*-values detected were 12 for PDT#3, and 0.3 for PDT, (attached excel named “*PDT alignment file.xlsx*”), [both for the protein Crf1, a transcriptional co-repressor of ribosomal genes, localizing to nucleus and cytoplasm <https://www.yeastgenome.org/locus/S000002631>]. Despite the relatively high *E*-values, we conducted individual alignments for the first 10 hits for PDT#3 using the Clustal Omega multiple sequence alignment platform, to investigate whether the PDT sequence aligns close to the C-terminus of these hits (attached file named “*PDT#3 alignment RefSeq hits.pdf*”). This analysis showed that none of the proteins showed similarity to the PDT#3 sequence, especially near the C-terminus.

The mitochondrial genome encodes 8 proteins for the OXPHOS subunits. Although none of them appeared as blast hits, we performed an alignment analysis to detect any sequences similar to PDT#3. Furthermore, given that Cox2 steady-state levels are downregulated in mf-Lon expressing strains (Figures 4b and Extended Figure 5c-d), all subunits of cytochrome c

oxidase (both mitochondrially- and nuclear-encoded) were also included in the alignment analysis (file name “*Alignment of mitoproteome with PDT#3.pdf*”). Our results revealed no protein with a reasonable sequence similarity with PDT#3 (or PDT) that could be assigned as a target for mf-Lon.

Nevertheless, it is noteworthy that *P_{ADHI}*-mito-mfLon has an impact on yeast nucleoid morphology and respiratory growth, implying that the protease may target non-tagged mitochondrial proteins (see Extended Data Figure 3). This effect was mitigated by reducing mf-Lon expression through the replacement of *PIMI* with mf-Lon, and therefore the latter strain is an appropriate background to study mitochondria-specific protein degradation. Further details can be found in the Results section under the heading "**Optimizing mf-Lon expression for a minimal impact on mtDNA and nucleoid structure.**"

We have now included a discussion on these points in the revised manuscript (lines 234-245).

4. “Moreover, the observation that Lon-induced degradation depends on the growth conditions (requiring adenine depletion after diauxic shift from fermentation to respiration) limits the potential of the outlined workflow. E.g. it is not conceivable that the system can be transferred to higher eukaryotes”

We now show that mf-Lon indeed degrades mitochondrially-targeted BFP-PDT in human cells, showing that the system has a broader applicability than yeast adenine auxotrophic mutants (Figure 5). Potentially, this indicates that adenine depletion of the yeast mutants creates a mitochondrial environment more like that present in human cells. As stated, this will be an interesting future topic of investigations.

5. “Hence, the proof-of-concept experiment shown in Fig. 4 targeting two proteins with known dual localisation may be extended by further analysing the specific function of the mitochondrial pool of the selected candidates in order to showcase the power of the system in combination with yeast as a model organism”

To show the power of the system we first performed PDT-degradation of Mip1, which is a protein exclusively localized to mitochondria, with a known function in mtDNA copy number maintenance⁹. Our mtDNA quantification showed that mf-Lon induced degradation of Mip1-PDT resulted in a strong depletion of mtDNA, phenocopying a *mip1Δ* strain

(Extended Data Figure 6a), and the inability of the cells to grow on respiratory medium (Extended Data Figure 6b). This demonstrates that site-specific degradation of mitochondrial proteins by mf-Lon has functional consequences.

Since *Apn1* and *Pif1* also function in the nucleus, their mitochondria-specific phenotypes have not been tested until now. Our results showed that mf-Lon-induced degradation of *Apn1*-PDT and *Pif1*-PDT did not substantially impact mtDNA levels (Extended Data Figure 6a). This aligns with the previously reported data which shows that the impact of *apn1* Δ and *pif1* Δ on mtDNA copy number is typically subtle, and evident only under genotoxic stress⁶⁻⁸.

The above-mentioned results have been added to the revised manuscript (lines 179-193) and are shown in Extended Data Figure 6.

Reviewer #3 (Remarks to the Author):

Sanyal and colleagues present an interesting approach to selectively deplete a mitochondrial pool of specific proteins. Such technique is interesting when the function of the mitochondrial fraction of a protein with a dual localization is analyzed. The technique is based on the expression of a Lon protease from the bacterium *Mesoplasma florum* in baker's yeast. Mf-Lon degrades proteins with a *ssrA* tag, termed PDT. Depletion of adenine in the growth media induces the degradation of PDT-tagged proteins inside mitochondria by mfLon. The experimental approach is interesting to analyze the functions of dually localized proteins. There are some concerns that the authors should address.

1. The Lon protease degrades many mitochondrial proteins. Does the expression of an additional Lon protease affect mitochondrial proteostases and cause pleiotropic effects. The authors should show steady state levels of stress marker proteins and show a growth test of the used strains.

We investigated the steady-state levels of the stress marker Hsp60 in Lon-expressing cells subjected to adenine depletion-mediated induction conditions. No upregulation of Hsp60 was detected (Extended Data Figure 3i), which would be expected in case of altered mitochondrial proteostasis¹⁰. This shows that expression of mf-Lon does not induce mitochondrial protein stress in the tested conditions. We next looked at the growth of the strains used, on respiratory and non-respiratory media. It revealed that mf-Lon expressed

from the *ADHI* promoter leads to a growth defect in respiratory medium, suggesting that prolonged overexpression of mf-Lon has a negative effect on mitochondrial function (Extended Data Fig 6b). Corroborating this, the mtDNA morphology looked slightly abnormal in these cells (Extended Data Figure 3a-f). These defects, however, were alleviated in the strain where we expressed mf-Lon from the *PIMI* locus, suggesting that reduced mf-Lon expression recapitulates a wild type-like situation (Extended Data Figure 3 and 6). We therefore believe that the *pim1Δ::mito-mf-Lon* background provides a better system to study mitochondria-specific protein degradation, which is now stated in the manuscript (lines 234-245).

2. The authors should also show that mfLon localizes to mitochondria.

We have visualized the localization of mf-Lon by C-terminus mCherry tagging and confirmed that it localizes to the mitochondria (Fig 3). Our mitochondrial preparations also show that the mf-Lon localizes to the mitochondria (Fig 4, Extended Data Fig 5). Furthermore, when we treated mitochondria isolated from mf-Lon expressing cells with proteinase K and triton X, we confirmed that the mf-Lon is localized in the mitochondrial matrix (shown below, not shown in the manuscript).

Figure 1. Mitochondria isolated from strains expressing *P_{ADHI}-mito-mf-Lon* grown overnight in minimal medium were treated with proteinase K, in isotonic (crude mitochondria, removing extra-mitochondrial proteins), or hypotonic condition (mitoplast, removing proteins present in the intermembrane space), or isotonic condition along with triton X (removing matrix localized proteins).

3. The authors should comment on why mfLon is not induced on full medium.

The exact reason for why PDT degradation is not induced in rich medium remains unclear. Depletion of adenine in the minimal medium appears, however, pivotal for mf-Lon mediated degradation of PDT-tagged proteins. The degradation of GFP-PDT is entirely reliant on the adenine mutation, as we did not observe PDT degradation in *ADE2*-proficient strains (data not presented). Yeast strains harbouring the *ade2* mutation are unable to synthesize P-ribosylaminoimidazolecarboxylate— a product required for adenine *de novo* synthesis – and accumulate oxidized ribosylaminoimidazole (manifested by a red pigment) when adenine is depleted ¹. Intriguingly, this red pigment accumulation depends on mitochondrial function, and *ade2* mutants have been widely used in screens for respiratory function through red/white colony screening ¹. Despite these observations, the molecular mechanism that links adenine and mitochondrial function remains unclear.

Significantly, we noticed that following adenine depletion from the culture medium, the mitochondrial network becomes more extensive, resembling the mitochondria in cells cultivated in respiratory conditions (see Figure 3 and reference ²). This could indicate that adenine depletion enhances the respiration of *ade2*-deficient yeast strains, thereby creating a mitochondrial environment that allows mf-Lon-dependent PDT degradation. However, this hypothesis requires further investigation, and in the light of our new results showing that mitochondria-specific mf-Lon degradation functions in human cells, we find that such analysis lies outside the scope of this investigation.

We have now included a discussion on these points in the revised manuscript (lines 246-264).

4. Figure 1B: Western blot would be important to reveal the depletion of the PDT-tagged proteins.

We have now added the Western blot in Extended Data Figure 1c.

5. The authors present that mitochondrial Apn1-PDT and Pif1-PDT are degraded by the used approach. To show the selective depletion of the mitochondrial pool the authors should show that the nuclear fractions of the proteins unaffected.

The nuclear fraction of Apn1-PDT and Pif1-PDT in Lon expressing strains has not been directly tested. However, we think that their extra-mitochondrial fractions are intact owing to the following observations: first, the mitochondrial specificity of mf-Lon-PDT degradation

can be demonstrated by our cytoplasmically-localized GFP-PDT, which showed that the mf-Lon caused PDT degradation exclusively within the mitochondria, and not outside (Fig 1D and E); second, mf-Lon does not significantly reduce Apn1-PDT levels in whole cell extracts but completely depletes Apn1-PDT within the mitochondrial fraction (Extended Data Figure 5c). This indicates that extra-mitochondrial Apn1-PDT remains unaffected and does not undergo PDT degradation outside of the mitochondria. Therefore, mf-Lon specifically degrades PDT-tagged proteins inside the mitochondria.

We have now included a discussion on these points in the revised manuscript (lines 225-233).

6. The authors state that they used proteinase K to remove nuclear contamination from the mitochondrial isolation. However, proteinase K treatment is typically used to remove non-imported proteins from the mitochondrial surface, but not entire organelles.

Thanks for the comment, we have now corrected the sentence (lines 169-170).

7. Furthermore, also Cox2 is reduced after proteinase K treatment (Figure E5c), indicating that the mitochondria are not fully intact. The authors should comment on this.

The apparent proteinase K-dependent decrease of Cox 2 in Figure E5c stems from less loading in lane 7 as compared to 5.

8. Is this approach specific for yeast cells or could it also be used in mammalian cell lines?

We now show that mf-Lon indeed degrades mitochondrially-targeted BFP-PDT in human cells, showing that the system has a broader applicability than yeast adenine auxotrophic mutants (Figure 5). Potentially, this indicates that adenine depletion of the yeast mutants creates a mitochondrial environment more like that present in human cells. As stated, this will be an interesting future topic of investigations.

REFERENCE:

1. Kim, G., Sikder, H. & Singh, K. K. A colony color method identifies the vulnerability of mitochondria to oxidative damage. *Mutagenesis* **17**, 375–381 (2002).
2. Bagamery, L. E., Justman, Q. A., Garner, E. C. & Murray, A. W. A Putative Bet-Hedging Strategy Buffers Budding Yeast against Environmental Instability. *Curr Biol* (2020) doi:10.1016/j.cub.2020.08.092.
3. Grilly, C., Stricker, J., Pang, W. L., Bennett, M. R. & Hasty, J. A synthetic gene network for tuning protein degradation in *Saccharomyces cerevisiae*. *Mol Syst Biol* **3**, 127 (2007).
4. Gur, E. & Sauer, R. T. Evolution of the *ssrA* degradation tag in *Mycoplasma*: specificity switch to a different protease. *Proc Natl Acad Sci U S A* **105**, 16113–16118 (2008).
5. Altschul, S. F. *et al.* Gapped BLAST and PSI-BLAST: a new generation of protein database search programs. *Nucleic Acids Res.* **25**, 3389–3402 (1997).
6. Vongsamphanh, R., Fortier, P.-K. & Ramotar, D. Pir1p Mediates Translocation of the Yeast Apn1p Endonuclease into the Mitochondria To Maintain Genomic Stability. *Mol Cell Biol* **21**, 1647–1655 (2001).
7. Acevedo-Torres, K., Fonseca-Williams, S., Ayala-Torres, S. & Torres-Ramos, C. A. Requirement of the *Saccharomyces cerevisiae* APN1 gene for the repair of mitochondrial DNA alkylation damage. *Environ. Mol. Mutagen.* **50**, 317–327 (2009).
8. Cheng, X., Qin, Y. & Ivessa, A. S. Loss of mitochondrial DNA under genotoxic stress conditions in the absence of the yeast DNA helicase Pif1p occurs independently of the DNA helicase Rrm3p. *Mol Genet Genomics* **281**, 635–645 (2009).
9. Genga, A., Bianchi, L. & Foury, F. A nuclear mutant of *Saccharomyces cerevisiae* deficient in mitochondrial DNA replication and polymerase activity. *J. Biol. Chem.* **261**, 9328–32 (1986).
10. Rao, K. B. N. *et al.* Stress Responses Elicited by Misfolded Proteins Targeted to Mitochondria. *J. Mol. Biol.* **434**, 167618 (2022).

The following are the bioinformatic analyses performed in response to the third question asked by Reviewer #2. There are 3 attachments in total:

1. *PDT alignment file.xlsx* (attached separately)
2. *PDT#3 alignment RefSeq hits.pdf* (attached below)
3. *Alignment of mitoproteome with PDT#3.pdf* (attached below)

REVIEWERS' COMMENTS

Reviewer #1 (Remarks to the Author):

The authors have convincingly responded to all previous queries. This reviewer is particularly pleased with the experiments showing that mf-Lon degrades mitochondrially-targeted BFP-PDT in human cells. This portrays the broad applicability of the described approach. The manuscript is now suitable for publication in Nat Comm.

Reviewer #2 (Remarks to the Author):

The authors have addressed all critical points.

Reviewer #3 (Remarks to the Author):

The authors addressed most of my concerns. The presented technique is interesting and represents an important tool to investigate dually localized proteins.

I have one minor comment:

The authors added data to show that their approach can also be used in human cells. However, the reduction of PDT-BFP is only moderate after LON induction as determined by the fluorescence signal. It could help to show western blots to clearly assess the reduction of the protein.

AUTHOR REBUTTAL (FINAL)

We thank all the reviewers for their comments and critical analysis of our work. We truly think that their feedback has immensely improved the quality of our manuscript. Please find our response to any remaining comments below:

Reviewer #1 (Remarks to the Author):

The authors have convincingly responded to all previous queries. This reviewer is particularly pleased with the experiments showing that mf-Lon degrades mitochondrially-targeted BFP-PDT in human cells. This portrays the broad applicability of the described approach. The manuscript is now suitable for publication in Nat Comm.

We are also very excited about this finding! We thank the reviewer for their comments.

Reviewer #2 (Remarks to the Author):

The authors have addressed all critical points.

We are pleased to have successfully done so.

Reviewer #3 (Remarks to the Author):

- 1. The authors addressed most of my concerns. The presented technique is interesting and represents an important tool to investigate dually localized proteins.**

We are happy that we were able to address most of the reviewer's concerns.

- 2. I have one minor comment:**

The authors added data to show that their approach can also be used in human cells. However, the reduction of PDT-BFP is only moderate after LON induction as determined by the fluorescence signal. It could help to show western blots to clearly assess the reduction of the protein.

Although it may seem that the Lon-induced BFP-PDT degradation is moderate from the fluorescent images, we are confident that the observed results reflect true degradation by mf-Lon, as we have clear evidence that the degradation is dose-dependent (Figure 5c-e), similarly to yeast. This also implies that the degradation by mf-Lon can be tuned, which will be important to investigate the depletion of proteins according to expression levels of target proteins.